# A natural mouse model reveals genetic determinants of systemic capillary leak syndrome (Clarkson disease)

Abbas Raza[1,7], Zhihui Xie[2,7], Eunice C. Chan[2], Wei-Sheng Chen[2], Linda M. Scott[2], A. Robin Eisch[2], Dimitry N. Krementsov[3], Helene F. Rosenberg[4], Samir M. Parikh [5], Elizabeth P. Blankenhorn[6], Cory Teuscher[1]* & Kirk M. Druey[2]*

The systemic capillary leak syndrome (SCLS, Clarkson disease) is a disorder of unknown etiology characterized by recurrent episodes of vascular leakage of proteins and fluids into peripheral tissues, resulting in whole-body edema and hypotensive shock. The pathologic mechanisms and genetic basis for SCLS remain elusive. Here we identify an inbred mouse strain, SJL, which recapitulates cardinal features of SCLS, including susceptibility to histamine- and infection-triggered vascular leak. We named this trait "Histamine hypersensitivity" (Histh/*Histh*) and mapped it to Chromosome 6. *Histh* is syntenic to the genomic locus most strongly associated with SCLS in humans (3p25.3), revealing that the predisposition to develop vascular hyperpermeability has a strong genetic component conserved between humans and mice and providing a naturally occurring animal model for SCLS. Genetic analysis of *Histh* may reveal orthologous candidate genes that contribute not only to SCLS, but also to normal and dysregulated mechanisms underlying vascular barrier function more generally.

[1] Departments of Medicine and Pathology, University of Vermont School of Medicine, Burlington, VT 05405, USA. [2] Lung and Vascular Inflammation Section, Laboratory of Allergic Diseases, National Institute of Allergy and Infectious Diseases, NIH, Bethesda, MD 20892, USA. [3] Department of Biomedical and Health Sciences, University of Vermont School of Medicine, Burlington, VT 05405, USA. [4] Inflammation Immunobiology Section, Laboratory of Allergic Diseases, National Institute of Allergy and Infectious Diseases, NIH, Bethesda, MD 20892, USA. [5] Division of Nephrology and Department of Medicine, Beth Israel Deaconess Medical Center and Harvard Medical School, Boston, MA 02215, USA. [6] Department of Microbiology and Immunology, Drexel University College of Medicine, Philadelphia, PA 19129, USA. [7] These authors contributed equally: Abbas Raza, Zhihui Xie. *email: c.teuscher@uvm.edu; kdruey@niaid.nih.gov

The systemic capillary leak syndrome (SCLS, Clarkson disease) is a rare disease. There are currently fewer than 200 cases with a confirmed diagnosis worldwide although its prevalence is on the rise, likely due to increased awareness among physicians and the public[1]. SCLS is characterized by transient but potentially lethal episodes of diffuse vascular leakage. Complications of acute SCLS include shock, compartment syndrome, and multi-organ dysfunction[2]. The pathogenic mechanisms underlying SCLS are unknown, and consequently treatments have been developed primarily by trial and error. SCLS attacks are diagnosed based on the clinical triad of hypotension, elevated hematocrit, and hypoalbuminemia. SCLS flares are frequently preceded by respiratory viral and other infections, suggesting a role for inflammation in the induction of acute vascular leak[3].

During SCLS flares, transient spikes in circulating angiogenic proteins known to trigger vascular hyperpermeability (e.g., angio-poietin 2 (Angpt2) and vascular endothelial growth factor (VEGFA)) have been detected[4,5]. Additionally, sera from SCLS patients during episodes have been shown to impair microvascular endothelial cell (EC) barrier function, whereas convalescent sera from these same patients are functionally benign[4,6]. These results suggest that humoral factors present during disease flares are responsible for promoting vascular leak and systemic pathology.

Patients with SCLS routinely develop symptoms in mid-life, and they lack a family history of this disorder, both findings suggest that the genetic basis of disease is multifactorial and complex. Early studies of our initial SCLS patient cohort resulted in the identification of a small genetic interval, 3p25.3, as the highest-ranking candidate susceptibility locus ($p \sim 10^{-6}$) with an odds ratio of ~41[7]. Whole exome sequencing (WES) of a single patient with fatal SCLS revealed a potentially pathogenic loss of function mutation in the gene *ARHGAP5*, which encodes a known of a regulator of endothelial permeability (p190BRho-GAP)[8]. Notably, this mutation has not been detected in any other subjects with SCLS[9]. These results suggest that SCLS may be genetically heterogenous, which is yet another obstacle to a more definitive analysis of this rare disorder. An appropriate animal model could not only help delineate the role of genetic factors in SCLS, but also would serve as a pivotal tool for modeling gene–environment interactions in numerous, often life-threatening, disorders and diseases in which vascular hyperperme-ability has a central pathogenic function (e.g. systemic anaphylaxis, sepsis, Ebola virus, and dengue[10–12]).

Using publicly available mouse phenotype data, we identified a strain of mice, SJL/J (SJL), that uniquely and spontaneously displays the clinical features of SCLS—hypoalbuminemia, elevated hemato-crit, and hypotension[13]. Here, we investigated the feasibility of using SJL mice as a model to interrogate pathophysiological mechanisms of SCLS. Previous studies suggested that the SJL strain of mice is susceptible to systemic histamine, a canonical mediator of vascular hyperpermeability[14]. We report herein that mortality of SJL mice in response to administration of histamine was highly correlated with evidence of increased vascular leakage in a pattern similar to that reported in SCLS patients. Classical linkage studies revealed that a recessive locus in SJL mice controlling histamine-induced mortality mapped to a region on mouse Chr6, which we designated *Histh* (histamine hypersensitivity). Strikingly, *Histh* is syntenic with human 3p25.3, the highest ranking SCLS susceptibility locus. Considering the similarity of the *Histh*-mediated phenotype to SCLS, the results suggest that humans and mice share genetic traits that pre-dispose both species to stress-induced vascular dysregulation.

## Results

### Dermal vasculature of SCLS patients is hyper-responsive to leak provocateurs.
In vitro studies of ECs isolated from skin of an SCLS patient demonstrated exaggerated responses to inflammatory mediators, suggesting that primary endothelial dysfunction con-tributes to the clinical symptoms of SCLS[8]. To test this hypothesis directly in situ, we injected histamine or morphine intradermally in patients with SCLS and healthy controls and measured the area of drug-induced skin wheals caused by fluid extravasation. Histamine evokes vascular leakage by acting directly on the endothelium whereas morphine functions indirectly through mast cell degra-nulation and release of various permeability-inducing mediators including histamine, leukotrienes, and prostaglandins; both agents have been used safely in a prior human study of cutaneous vascular responsiveness[10,15,16]. We observed larger wheal sizes in SCLS patients compared to healthy controls in response to a range of concentrations of either histamine or morphine (Fig. 1a, b). Thus, with two unrelated stimuli provoking exaggerated vascular leakage in SCLS patients—and doing so in a dose-proportional fashion—the results were most suggestive of a generalized vascular hyper-responsiveness in SCLS.

### SJL mice exhibit traits that phenocopy human SCLS.
Seeking an in vivo model of vascular hyper-responsiveness, we first identified

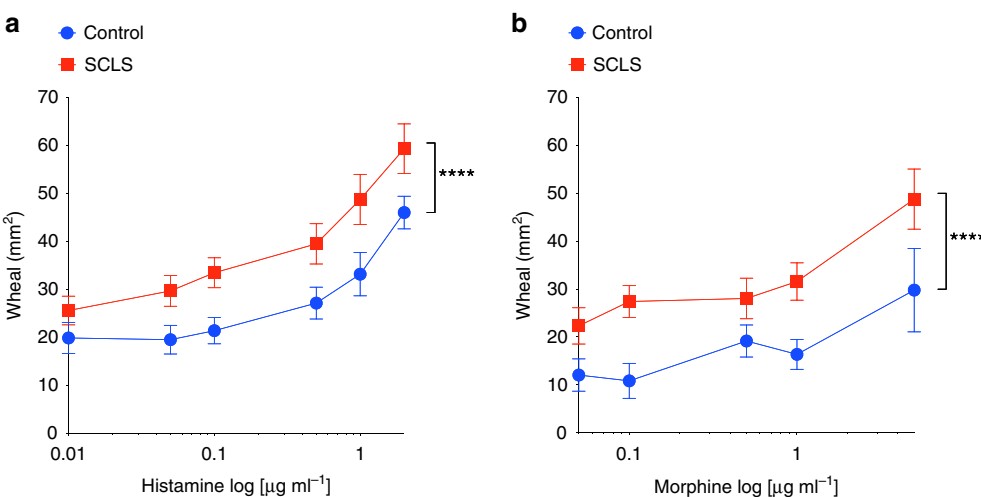

**Fig. 1** SCLS patients are hyper-responsive to histamine and morphine. **a, b** SCLS patients ($n = 16$) or healthy controls ($n = 7$) were injected intradermally with the indicated concentrations of histamine (**a**) or morphine (**b**) (log scale). Wheal sizes were determined using ImageJ. ****$p < 0.00001$, 2-way ANOVA

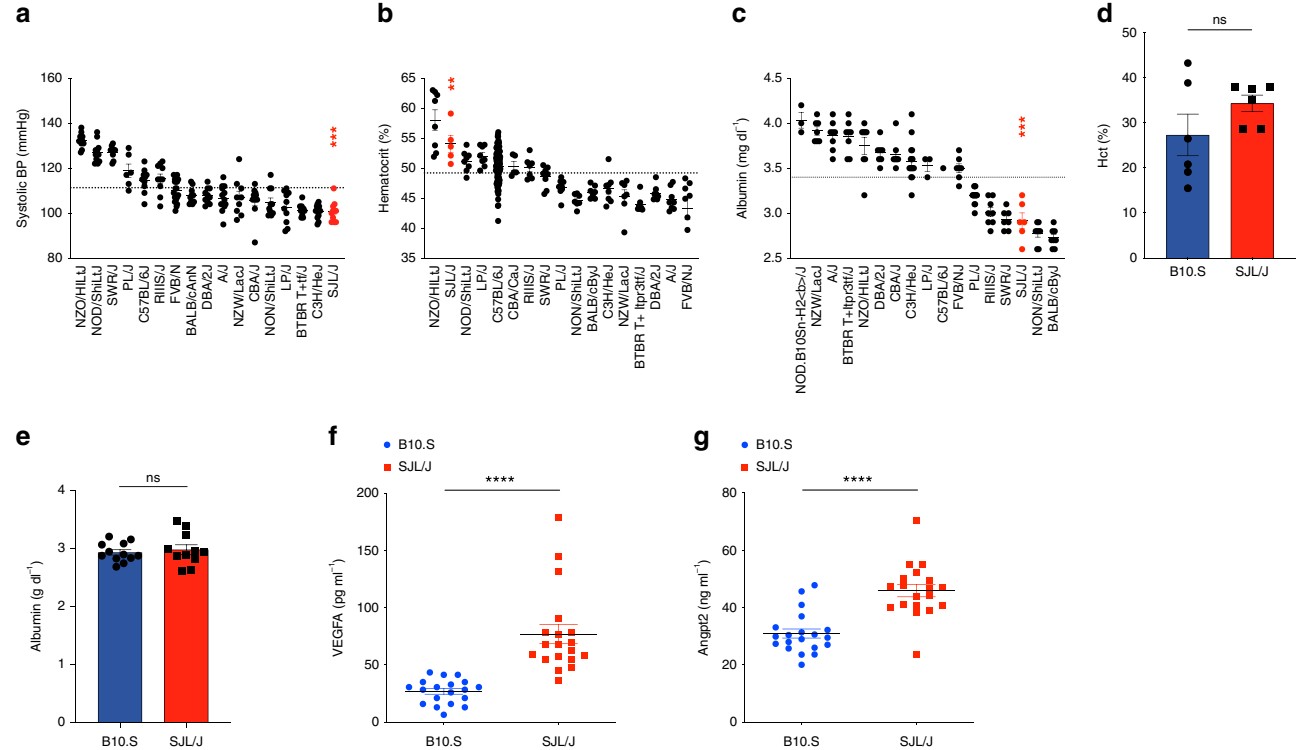

**Fig. 2 SJL/J mice phenocopy SCLS. a–c** Phenotype data were obtained from mouse phenome database (https://phenome.jax.org/) for systolic blood pressure (a, MPD#23602), hematocrit (b, MPD#31825) and plasma albumin (c, MPD#24451). Values for SJL/J mice (red) were compared each mean trait variable for all strains (**$p = 0.006$, ***$p = 0.0001$, Mann–Whitney U test). Bar graphs show mean ± s.d. values of hematocrit (**d**), serum albumin (**e**), serum levels of VEGFA (**f**) or Angpt2 (**g**) in B10.S and SJL/J mice (ns not significant, ****$p < 0.0001$, Mann–Whitney)

mouse strains with a constellation of traits resembling SCLS by searching the Mouse Phenome Database (https://phenome.jax.org) for data on systolic blood pressure, hematocrit, and plasma albumin. Compared to the mean trait variables for all strains, SJL mice were unique in that they exhibited significantly lower systolic blood pressure in conjunction with an increased hematocrit and hypoalbuminemia ($p < 0.01$, 2-way ANOVA, SJL v. all other strains (Fig. 2a–c). As a comparator strain for functional studies, we selected the B10.S/SgMcdJ (B10.S) mouse, which, like SJL, carries the $H2^S$ haplotype at the $H2$ (MHC) locus, but has been reported to be insensitive to histamine[14]. We have previously used various SJL/B10.S crosses in genetic studies of susceptibility to autoimmune neuroinflammation[17], a disease that may be regulated in part by histamine-mediated effects on vascular or immune systems, and controlling for MHC-regulated effects in studies of immune/inflammatory disorders is paramount[18,19]. However, phenotypic data were not available for B10.S mice or C57BL/10SgSnJ (the background recipient sub-strain used to generate B10.S) mice, and blood pressure and Hct values reported in Fig. 2a–c were in large part obtained from young mice (<6 months of age). Because SCLS most often commences in middle-aged adults, we instead measured serum Hct and albumin directly in aged (>6 months of age) B10.S and SJL mice. At homeostasis, Hct and serum albumin values in these two strains were equivalent and within the normal range of the testing laboratory (Fig. 2d, e). Unfortunately, we were unable to obtain accurate blood pressure measurements in our laboratory. Similar to SCLS patients[4], SJL mice also exhibited increased circulating levels of SCLS-related angiogenic proteins Angpt2 and VEGFA relative to B10.S mice (Fig. 2f–g), results suggesting that they might be more susceptible to vascular leakage. Importantly, asymptomatic SCLS patients have normal Hct, albumin, and

blood pressure during convalescent intervals[1]. Taken together, our findings therefore suggest that SJL mice are useful as an in vivo model for SCLS and that B10.S mice are suitable for comparative functional studies of vascular leakage.

**SJL mice exhibit age- and inflammation-dependent hypersensitivity to histamine.** The SJL strain has been extensively studied, most notably to investigate immune dysregulation[20]. To the best of our knowledge, however, vascular function in SJL mice has not been queried in detail. To determine whether SJL mice were more susceptible to vascular leakage than B10.S mice, we administered histamine to two distinct SJL sub-strains, SJL/J and SJL/NCr mice, with B10.S mice as a control. Both SJL sub-strains exhibited histamine hypersensitivity and died within 30 min of histamine administration, even at the lowest intravenous doses, whereas B10.S mice all survived (Table 1a). We have designated this phenotype "histamine hypersensitivity (Histh)". Similar to the emergence of SCLS symptoms in middle age[1], we found that the Histh phenotype in SJL mice was age-dependent; mice greater than 6 months of age all succumbed to histamine administration whereas the younger, 8-week-old mice did not (Table 1a).

In a recent survey of SCLS patients, infectious triggers were identified in 35–50% of disease exacerbations[3]. We therefore tested whether an inflammatory stimulus potentiates a lethal response to histamine challenge in SJL mice. To study this, we administered complete Freund's adjuvant (CFA), a complex mixture of antigens and oil widely used to augment immune responses. CFA-induced inflammation had no apparent impact on the responses of aged (>6 months) mice; both primed and un-primed mice succumbed to intravenous histamine doses at 25 mg/kg and higher, but not to the lowermost dose (12.5 mg/kg).

**Table 1 SJL mice exhibit age and/or inflammation-dependent histamine hypersensitivity (Histh)**

| A | | | | B | | | |
|---|---|---|---|---|---|---|---|
| Strain | Histamine (mg/kg) | aged | 8 week | Strain | Histamine (mg/kg) | CFA aged | 8 week |
| SJL/J | 100 | 4/4 | 0/4 | SJL/J | 100 | 4/4 | 4/4 |
| | 50 | 4/4 | 0/4 | | 50 | 4/4 | 4/4 |
| | 25 | 2/4 | 0/4 | | 25 | 4/4 | 4/4 |
| | 12.5 | 0/4 | 0/4 | | 12.5 | 0/4 | 0/4 |
| SJL/NCr | 100 | 4/4 | 0/4 | SJL/NCr | 100 | 4/4 | 4/4 |
| | 50 | 4/4 | 0/4 | | 50 | 4/4 | 4/4 |
| | 25 | 2/4 | 0/4 | | 25 | 4/4 | 4/4 |
| | 12.5 | 0/4 | 0/4 | | 12.5 | 0/4 | 0/4 |
| B10.S/SgMcdJ | 100 | 0/4 | 0/4 | B10.S/SgMcdJ | 100 | 0/4 | 0/4 |
| | 50 | 0/4 | 0/4 | | 50 | 0/4 | 0/4 |
| | 25 | 0/4 | 0/4 | | 25 | 0/4 | 0/4 |
| | 12.5 | 0/4 | 0/4 | | 12.5 | 0/4 | 0/4 |
| (B10.S x SJL) F$_1$ | 100 | 0/4 | 0/4 | (B10.S x SJL) F$_1$ | 100 | 0/4 | 0/4 |
| | 50 | 0/4 | 0/4 | | 50 | 0/4 | 0/4 |
| | 25 | 0/4 | 0/4 | | 25 | 0/4 | 0/4 |
| | 12.5 | 0/4 | 0/4 | | 12.5 | 0/4 | 0/4 |

Young (8–10-week old) or aged (>6 months) mice were left untreated or pretreated with CFA by intraperitoneal (i.p.) injection and challenged 30 days later with the indicated doses of histamine (mg/kg) by i.v. injection. Deaths were recorded at $t = 30$ min. Results are expressed as the number of deaths/total mice
*CFA* complete Freund's adjuvant

In contrast, the younger (8-week old) SJL mice, which were fully resistant to intravenous histamine alone, exhibited 100% mortality in response to histamine (25 mg/kg and higher doses) if first primed with CFA. CFA-primed B10.S mice of both age groups remained resistant throughout. Furthermore, (B10.S × SJL) F1 hybrid mice phenocopied B10.S mice, demonstrating that Histh is a recessive trait (Table 1b). Taken together, these findings suggest that genetically controlled histamine hypersensitivity can be spontaneous and/or exacerbated by inflammatory stimuli.

Finally, we confirmed that histamine elicited death of SJL mice in a manner consistent with the published literature[21,22]. Similar to SCLS in humans, there was a rapid onset of hemoconcentration and hypovolemic shock within 5–10 min of histamine administration. In both SJL and B10.S mice there was an increase in Hct over baseline; however, Hct values were significantly higher in SJL mice than in B10.S mice after histamine administration (~66% vs. 57%, $p = 0.02$) (Fig. 3a). In contrast, serum albumin values were normal and equivalent in both strains prior to and immediately after histamine administration (Fig. 3b). This finding is consistent with the presentation of SCLS flares, in which serum albumin levels are typically normal at initial presentation, followed by a gradual decrease over the following 24–36 h[23]. In further accordance with SCLS in humans, mice examined immediately after death had unobstructed lungs, a small, non-dilated heart, an uncongested liver, and grossly normal kidneys and intestines. Likewise, the histological appearance of heart, liver, kidneys, and small intestines was essentially normal in both SJL and B10.S mice (Fig. 3c). The lungs also appeared to be normal except for the presence of dense peribronchial and perivascular lymphoid aggregates in some SJL mice. We suspect that these represented reticulum cell tumors, which have previously been reported to develop in aged (greater than 6 months of age) SJL mice[24]. However, these abnormalities were also detected in lungs of untreated mice and were thus unrelated to histamine administration (Supplementary Fig. 1). Taken together, our findings suggest that histamine caused death of SJL mice by inducing massive fluid extravasation, resulting in the inability to compensate sufficiently to maintain blood pressure and venous return to the heart, a phenotype which reflects the acute presentation of SCLS attacks in humans.

**Histh, the locus controlling susceptibility to vascular hypersensitivity to histamine, exhibits maximal linkage to mouse chromosome 6.** To map the gene or genes controlling Histh, we treated ~478 (B10.S × SJL) F2 mice with histamine at 30 days after priming with CFA and performed genetic association analysis using pre-established genomic markers[25] (Table 2). A genome scan using microsatellite markers that distinguish Histh-resistant B10.S and Histh-susceptible SJL mice identified a quantitative trait locus (QTL) on Chr6 within an approximately ~100 Mb region between *D6Mit74*(48.72 Mb) to *D6Mit372*(148.45 Mb) ($p = 5.73 \times 10^{-5}$). In addition, there were minor linkages to Chr8 ($p = 2.80 \times 10^{-2}$) and Chr15 ($p = 9.74 \times 10^{-4}$, Supplementary Data 1). We have designated this locus on Chr6 as *Histh* (histamine hypersensitivity).

**Congenic mapping of Histh.** We then confirmed the existence and location of Histh on Chr6 by congenic mapping (Table 3). We used marker-assisted selection to introgress the Histh interval (*D6Mit74* (48.72 Mb) through *D6Mit254* (125.36 Mb) from SJL onto the B10.S background. These mice were backcrossed for 12 generations and fixed as a homozygous interval-specific recombinant congenic line (ISRCL) hereafter referred to as B10.S-*Histh*$^{SJL}$. The Histh phenotype was confirmed by testing susceptibility to histamine challenge 30 days after priming with CFA as above. Indeed, lethality due to Histh differed significantly among the strains ($X^2 = 51.61$, df = 1, $p < 0.0001$); SJL and B10.S-*Histh*$^{SJL}$ mice were significantly more susceptible to CFA/histamine than were B10.S mice ($X^2 = 55.24$, df = 1, $p < 0.0001$ for both strains), but their responses did not differ significantly from each other. Moreover, (B10.S × B10.S-*Histh*$^{SJL}$) F1 hybrids were Histh-resistant, confirming the observation made earlier (see findings in Table 1) regarding Histh as a recessive trait. Thus, we have physically mapped Histh to Chr6:48–125 Mb and demonstrated that this locus is sufficient to provide full penetrance of the Histh phenotype.

**Dermal vasculature of mice harboring an Histh susceptibility allele is hyper-responsive to histamine.** We hypothesized that Histh in mice is due to a genetic predisposition of ECs to exaggerated barrier breakdown in response to permeability mediators. To evaluate histamine-mediated vascular hyperpermeability

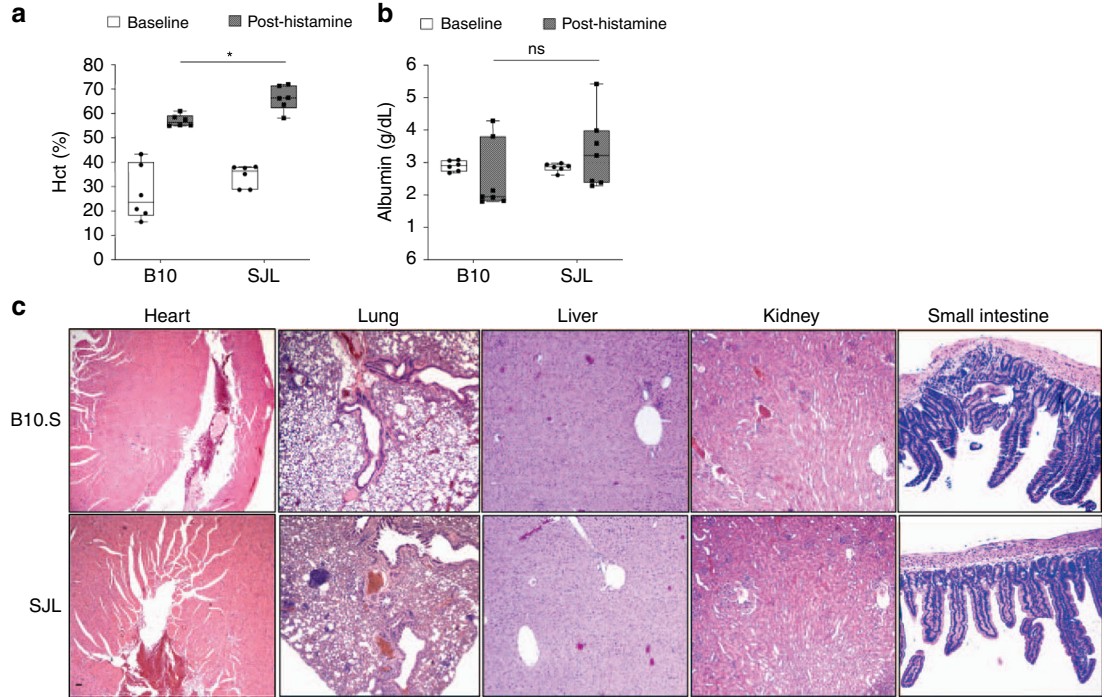

**Fig. 3** Phenotypic characterization of mouse strains used to model SCLS. **a, b** Hematocrit (a) or serum albumin (b) were measured in untreated SJL or B10.S mice. The mice were subsequently challenged with histamine systemically via the intraperitoneal route (25 mg/kg) and blood was withdrawn at $t = 10$ min followed by measurement of these parameters post-treatment. (mean ± s.e.m. values; each symbol represents an individual mouse *$p = 0.03$, two-way ANOVA, Sidak multiple comparisons). **c** Histology of organs from histamine-treated SJL or B10.S assed by hematoxylin and eosin. Images are representative of four mice in each group. Scale bar = 50 μm

| Marker | Location | Dead | | | Alive | | | $X^2$ | P-value |
|---|---|---|---|---|---|---|---|---|---|
| | | B10.S Ho | het | SJL Ho | B10.S Ho | het | SJL Ho | | |
| D6Mit116 | Chr6:25150229-25150350 | 29 | 36 | 53 | 90 | 71 | 203 | 6.84 | 3.28E-02 |
| D6Mit74 | Chr6:48726556-48726705 | 21 | 47 | 48 | 111 | 66 | 189 | 25.69 | 2.64E-06 |
| D6Mit17 | Chr6:71119218-71119467 | 16 | 56 | 45 | 116 | 57 | 180 | 51.16 | 7.76E-12 |
| D6Mit8 | Chr6:83713869-83714031 | 12 | 34 | 54 | 91 | 127 | 42 | 55.70 | 8.00E-13 |
| D6Mit178 | Chr6:94225829-94225955 | 13 | 31 | 56 | 94 | 130 | 37 | 67.80 | 2.00E-15 |
| D6Mit36 | Chr6:104503360-104503555 | 8 | 62 | 51 | 124 | 52 | 198 | 81.61 | 1.90E-18 |
| D6Mit54 | Chr6:112269957-112270141 | 5 | 38 | 55 | 89 | 137 | 35 | 80.30 | 3.70E-18 |
| D6Mit366 | Chr6:115242853-115249277 | 6 | 38 | 56 | 88 | 137 | 35 | 74.40 | 2.60E-17 |
| D6Mit216 | Chr6:121115242-121115387 | 9 | 65 | 41 | 120 | 42 | 192 | 103.36 | 3.59E-23 |
| D6Mit254 | Chr6:125356646-125356785 | 11 | 65 | 42 | 127 | 51 | 194 | 89.47 | 3.73E-20 |
| D6Mit59 | Chr6:138976326-138976494 | 10 | 58 | 49 | 124 | 56 | 188 | 66.44 | 3.74E-15 |
| D6Mit372 | Chr6:148450482-148450593 | 8 | 62 | 47 | 115 | 56 | 195 | 75.63 | 3.77E-17 |

**Table 2 Histamine sensitivity (Histh) maps to mouse chromosome 6**

(B10.S x SJL) F$_2$ mice were genotyped using microsatellite markers (21), and phenotyped for Histh: histamine sensitivity was determined by i.v. injection of 50 mg/kg histamine free base in 0.2 ml of PBS 30 days post-CFA injection. Deaths were recorded $t = 30$ min after histamine challenge

in vivo, we used the well-established Miles assay[26], which measures plasma extravasation from cutaneous microvasculature through quantification of Evans blue (EB), an albumin-binding dye, in skin. We first injected EB intravenously into older (>6 months) B10.S or SJL mice, followed by intradermal injection of histamine and quantification of extravasated EB in skin biopsies. Dermal EB extravasation increased significantly in skin biopsies of histamine-treated SJL v. B10.S mice, and extravasation in both strains appeared to be more extensive than that detected in response to PBS alone (Fig. 4a, b). To determine if the Histh locus is associated with histamine-induced vascular leakage, we performed the Miles assay in older B10.S-Histh$^{SJL}$ congenic mice. Compared with the responses of B10.S mice, B10.S-Histh$^{SJL}$ congenic mice exhibited an increase in dermal EB vascular leakage (Fig. 4c).

Finally, to determine if the increased susceptibility to histamine-induced cutaneous vascular leak in SJL and B10.S-Histh$^{SJL}$ mice is age-dependent and/or inflammation dependent, we performed Miles assays in 8-week-old mice that were primed with CFA prior to histamine challenge. In the absence of CFA priming, we observed no significant response to histamine in these younger mice (Fig. 4d). In contrast, CFA priming

**Table 3 Congenic mapping confirms the existence and location of the *Histh* locus**

| Strain | Marker/Location(bp) | | | | | | Histh | |
|---|---|---|---|---|---|---|---|---|
| | D6Mit74 48726556-48726705 | D6Mit17 71119218-71119467 | D6Mit178 94225829-94225955 | D6Mit54 112269957-112270141 | D6Mit254 125356646-125356785 | D6Mit372 148450482-148450593 | CFA | aged |
| SJL/J | S | S | S | S | S | S | 12/16 | 10/16 |
| B10.S | B | B | B | B | B | B | 1/118 | 0/16 |
| B10.S-*Histh*$^{SJL}$ | B | S | S | S | S | B | 34/73 | 4/8 |
| (B10.SxB10.S.*Histh*$^{SJL}$) F$_1$ | B | B/S | B/S | B/S | B/S | B | 0/15 | ND |

Cohorts of young (8–10-week old) mice pre-conditioned with CFA, or aged mice (>6 months) left untreated were challenged 30 days later with histamine (50-100 mg/kg) by i.v. injection. Deaths were recorded at t = 30 min. Results in the two right columns indicate the number of animals dead/total mice
*ND* not done, *CFA* complete Freund's adjuvant

potentiated the vascular hyperpermeability response in the younger 8-week-old B10.S-*Histh*$^{SJL}$ congenic mice. B10.S controls exhibited no increase in hypersensitivity to histamine following CFA administration. Together, these results strongly suggest that the *Histh* locus plays a critical role in regulating histamine-induced vascular hyperpermeability, and that this phenotype is affected by both age and pre-existing systemic inflammation.

**Susceptibility to histamine-induced systemic vascular leak is controlled by *Histh*.** For unknown reasons, vascular leak in SCLS patients manifests prominently in skin and skeletal muscle, less frequently in gastrointestinal tract and myocardium[27–29], and rarely in other internal organs including lungs, kidneys, and central nervous system[1,27]. To determine the extent of vascular leak in individual internal organs in response to histamine, we challenged young CFA-primed SJL, B10.S-*Histh*$^{SJL}$ and B10.S mice intravenously with EB followed by systemic (intraperitoneal) administration of histamine or diluent control; EB content was quantified in various organs after 30 min. Histamine-mediated vascular leak was detected in skin and skeletal muscle of both SJL and B10.S-*Histh*$^{SJL}$ but not B10.S mice compared to PBS-treated counterparts (Fig. 5a). We detected no dye extravasation in lungs, heart, or gut. We also observed a similar pattern of vascular leakage among older mice (greater than 6 months) following systemic administration of EB and histamine in the absence of CFA priming (Fig. 5b). These data indicate that the *Histh* locus controls susceptibility to histamine-mediated vascular hyperpermeability with impact in a whole animal model. Moreover, the pattern of vascular leakage is highly reminiscent of that observed in SCLS patients, where skin edema is profound and frequently complicated by extensive rhabdomyolysis requiring fasciotomies[1,27].

**Infectious triggers exacerbate genetically controlled vascular hyperpermeability.** Given the prominent link between viral upper respiratory tract or other infections and acute SCLS flares, we determined whether acute virus infection, as a common link and physiologic inflammatory stimulus, also elicits vascular leakage in SJL mice. We inoculated SJL and B10.S mice with influenza virus A (H3N2) and assessed systemic vascular leak in correlation with systemic symptoms (i.e. weight loss). A pronounced, 15–20% weight loss was apparent in both strains after 7 days of infection indicating comparable susceptibility to H3N2 (Fig. 6a). However, compared to uninfected controls at day 7 after infection, vascular leakage was increased in H3N2-infected SJL mice but not in B10.S mice (Fig. 6b). In line with the histamine-challenge results, EB extravasation was most prominent in skin, similar to the distribution of fluid extravasation in SCLS. These results demonstrate that a clinically relevant infectious trigger can exacerbate genetically controlled vascular hyperpermeability and suggest that the SJL mouse recapitulates multiple aspects of SCLS susceptibility, providing a useful and tractable animal model.

**Synteny of *Histh* locus and SCLS GWAS candidates.** The extreme rarity of SCLS has greatly limited our understanding of the complex genetic factors that contribute to disease development. The one and only published genome-wide association study of SCLS patients predicted genetic associations (653 SNPs, 139 genes) linked to disease including three SNPs in *CAV3* on Chr3p25.3 ($p \sim 10^{-6}$), with an odds ratio of ~41, as the highest-ranking susceptibility locus[7]. Considering the similarity between Histh and SCLS, we determined whether the *Histh* locus harbors any of the SCLS GWAS candidates. Synteny mapping revealed several human genes, including *CAV3*, *RAD18*, and *ATP2B2*, that were also captured in *Histh* (Fig. 7a). We generated protein functional interaction networks[30] of shared sub-phenotypes between Histh and SCLS to interrogate potential mechanistic links between *Histh*-associated genetic loci and disease (Fig. 7b). This approach identified several genes that are associated with aging (*ATP2B2*, *CAV3*, *CNTN3*, *CTNNA2*, *GRID2*), inflammation (*ATP2B2*, *CAV3*, *RAD18*, *KBTBD8*), vascular permeability (*SFXN5*, *RAD18*) and anaphylaxis (*CAV3*, *RAD18*, *CTNNA2*, *ATP2B2* and *GRID2*). In summary, these results suggest that SJL mice and human subjects with SCLS share a similar genetic basis for increased susceptibility to vascular hyperpermeability.

**Discussion**

SCLS is a unique, relapsing-remitting disease that can have devastating consequences. Although disorders with features of SCLS have recently emerged in children[31], most patients present in mid-life and lack any family history of the disease. Not unexpectedly, WES performed on DNA samples from several children with SCLS, their families, and unrelated adults did not uncover any shared single nucleotide variants that could readily explain the phenotype[9]. Thus, multiple genetic abnormalities may contribute to SCLS, indicating that our alternative approach of synteny studies may be more appropriate.

Our discovery of a shared susceptibility locus for vascular hyperpermeability in mice has led to unexpectedly strong conclusions about human 3p25 increasing the risk of SCLS in a mechanistic fashion. Specifically, we have characterized the vascular phenotype of the inbred SJL mouse strain, which shares genetic and phenotypic similarities with human patients with SCLS. Because SJL mice recapitulate cardinal features of SCLS, this mouse model may serve to advance our understanding of disease mechanisms. Just as patients with SCLS are typically

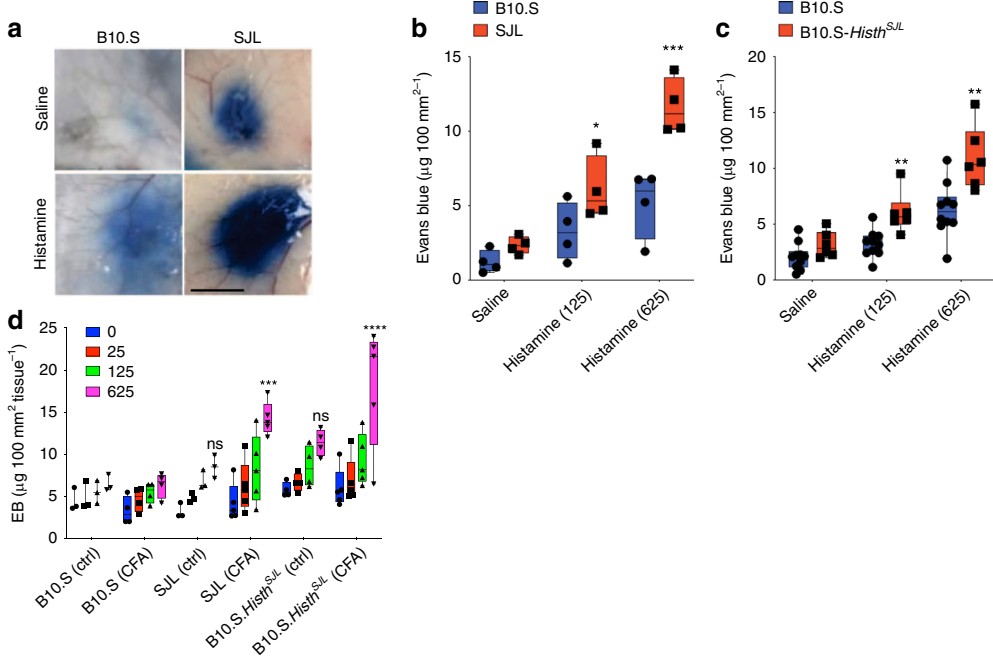

**Fig. 4** Dermal vasculature of mice containing the identified *Histh* allele is hyper-responsive to histamine. (**a-d**) SJL, B10.S or B10.S-*Histh*[SJL] congenic mice were injected with Evans Blue (EB) dye i.v. followed by intradermal challenge with histamine for 30 min. **a** Skin biopsies from aged (>6 months) mice after intradermal treatment with histamine (625 ng/mouse) or saline. Scale bar = 1 cm. **b, c** Quantification of EB extravasation in skin biopsies of B10.S, SJL, or B10.S-*Histh*[SJL] congenic mice. Each symbol represents one mouse; **$p < 0.003$; ****$p = 0.00005$; Holm-Sidak corrected $t$ test. **d** Young (8-week old) mice were primed with CFA or left untreated (ctrl) prior to histamine challenge. ***$p = 0.0006$; ****$p < 0.0001$, 2-way ANOVA, Tukey multiple comparisons v. B10.S mice ($n = 3-5$ mice/group) ns not significant

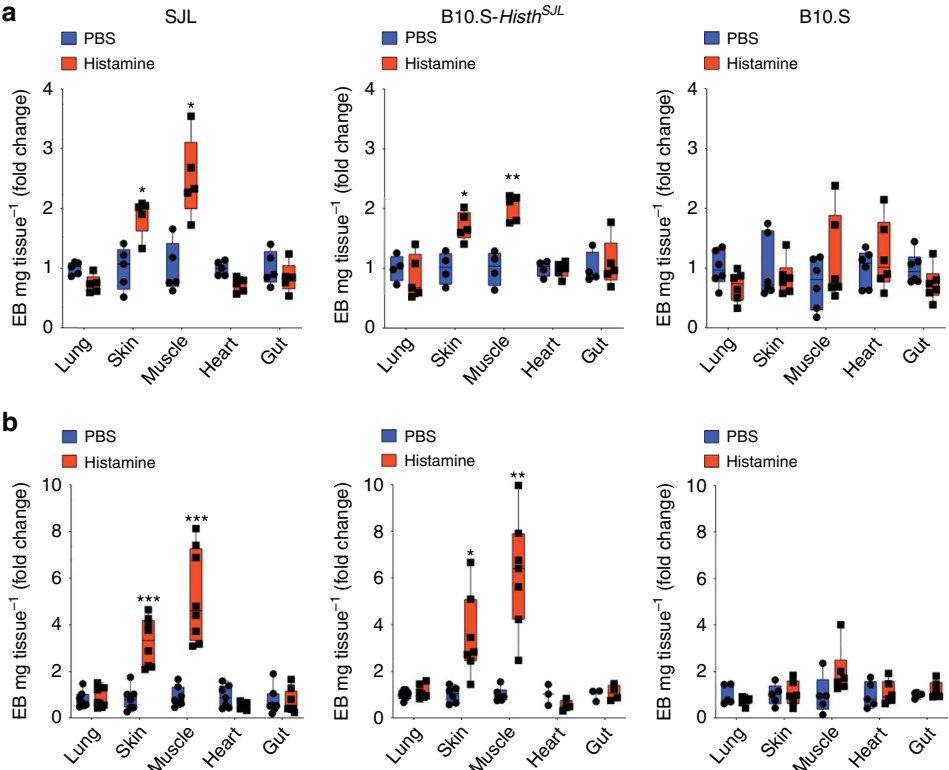

**Fig. 5** Systemic administration of histamine induces vascular leak in SJL and B10.S-*Histh*[SJL] but not B10.S mice. **a, b** EB was injected i.v. followed by i.p. injection of histamine (12.5 mg/kg) in either young (8-week old) mice primed with CFA (a) or aged mice (>6 months of age) (b). Extravasated dye was normalized to dry weight of the tissue/organ and expressed as fold change compared with controls (PBS). Each symbol represents an individual mouse; two independent experiments were performed; *$p < 0.03$, **$p = 0.001$; ***$p = 0.0004$, Holm-Sidak corrected $t$ test

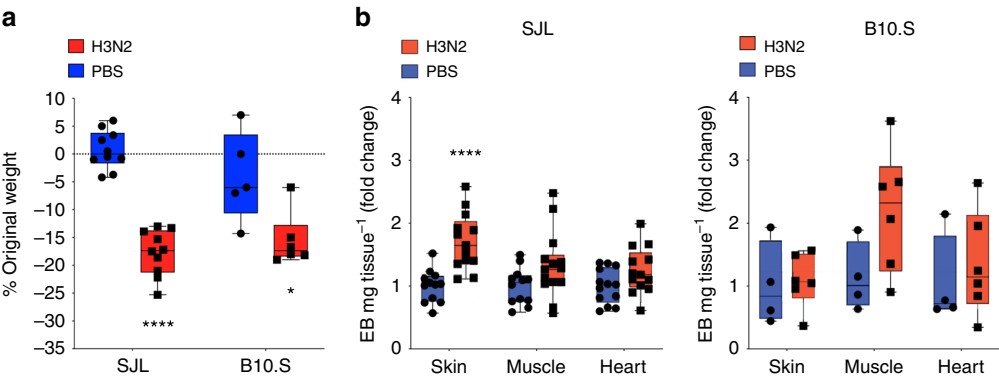

**Fig. 6** Influenza-associated vascular leak in SJL mice as a means to model SCLS. SJL and B10.S mice were infected with influenza virus A/H3N2. **a** Weights were determined day 0 and day 7 post-infection (*$p = 0.01$, ****$p < 0.00001$, Holm-Sidak corrected $t$ test. **b** EB dye extravasation was evaluated at day 8 post-infection; symbol represents an individual animal; 2–4 separate experiments were performed. ***$p = 0.0002$, Holm-Sidak corrected $t$ test

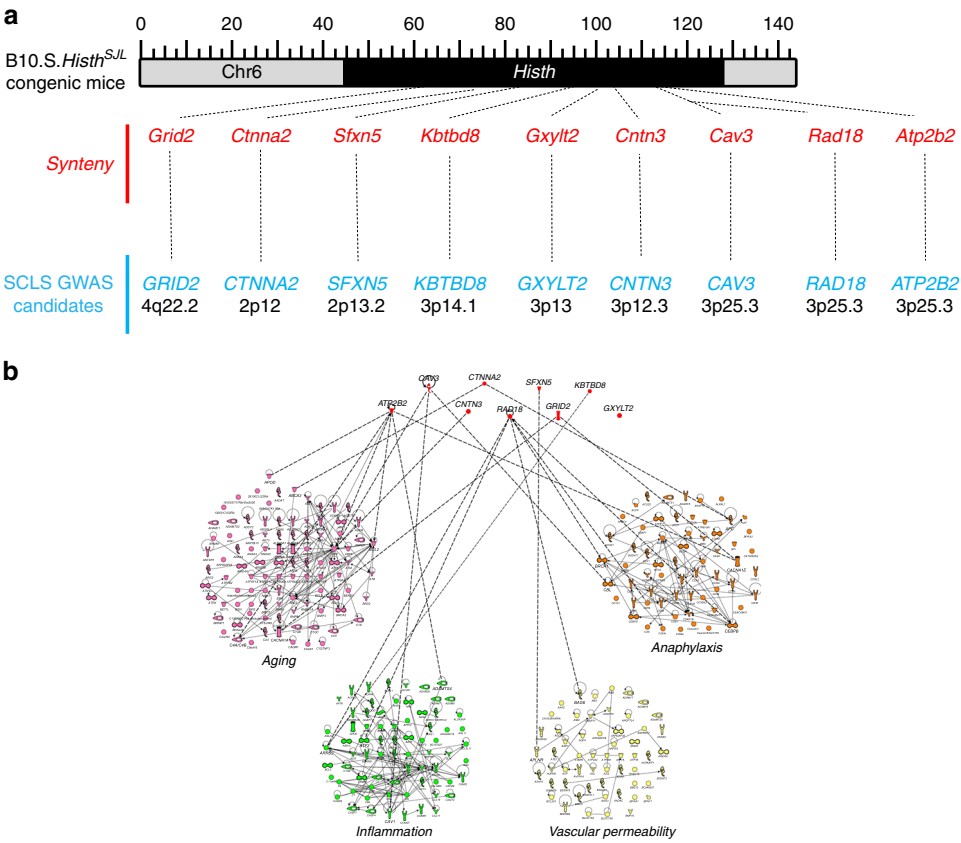

**Fig. 7** Shared genetic and phenotypic alignment between Histh and SCLS. **a** Of the 139 genetic loci implicated in SCLS (7), 9 (shown in blue) were found to overlap with the *Histh* locus on mouse Chr6 (shown in red). **b** Protein functional interaction networks for aging, inflammation, vascular permeability and anaphylaxis were generated using Ingenuity Pathway Analysis. These networks were used to assess biological interactions with 9 predicted SCLS candidates. Dotted lines represent an interaction. Each sub-phenotype and associated loci are color coded. Aging pink, Inflammation green, Vascular permeability yellow, Anaphylaxis orange

asymptomatic between episodes, SJL mice do not exhibit overt symptoms of vascular leak at baseline and have normal hematocrit and serum albumin at homeostasis (Fig. 1). Histamine-induced mortality in SJL mice correlated with vascular leakage in skin and skeletal muscle, which are the most prominent sites of pathology in SCLS patients. Both SCLS patients and SJL mice are uniquely susceptible to inflammation-associated vascular leakage, particularly that induced by systemic infection, such as that resulting from infection with influenza A.

Although we note that SJL mice have monoclonal gamma globulins in serum as do more than 80% of SCLS patients[1], this trait is controlled by the mammary tumor virus locus 29 (*Mtv29*), which encodes for an endogenous superantigen (vSAg29)[32]. Our genetic mapping studies exclude its role in increased histamine susceptibility as it falls outside of the *Histh* locus. Accordingly, no pathogenic role for SCLS paraproteins has been demonstrated thus far in humans[4,33]. Based on the histamine-induced mortality and vascular leakage findings, we can conclude that the *Histh*

locus controls histamine sensitivity and vascular permeability as a function of age and various inflammatory stimuli (CFA, viral infection). These patterns reflect the pathogenesis of SCLS, notably reflecting the fact that most patients who have spontaneous episodes are middle-aged. Interestingly, SCLS crises in children are nearly always preceded by infection[31].

Most important, the synteny map of the *Histh* locus and SCLS GWAS has provided strong focus on several gene candidates that may be involved in the pathogenesis of both Histh and SCLS in that they have demonstrated roles in processes involved in vascular endothelial barrier integrity. We further characterized potential functions of several of these genes in SCLS by searching PubMed and the International Mouse Phenotyping Consortium (http://www.mousephenotype.org) for associations with processes involved in vascular barrier integrity. Among these, *ATP2B2/Atp2b2* encodes ATPase plasma membrane $Ca^{2+}$ transporting 2 protein, which plays a critical role in intracellular calcium homeostasis and EC responses to histamine[34]. Atp2b2 also regulates endothelial nitric oxide (NO) synthase (eNOS) phosphorylation in ECs[35], a critical step in histamine- and VEGF-induced vascular permeability previously implicated in SCLS-associated vascular dysregulation[36,37]. Notably, $Atp2b2^{-/-}$ mice have reduced serum albumin levels at homeostasis compared to WT controls[38]. *Cav3* encodes caveolin 3, a protein also implicated in eNOS regulation[39]. Although *Cav3* knockout mice develop heart failure due to myocardial fibrosis and dilated cardiomyopathy[40], functions of Cav3 in the peripheral vasculature have not been studied in mice. *CTNNA2/Ctnna2* encodes α-catenin 2, which functions as a linker between cadherin adhesion receptors and the cytoskeleton, and thereby regulates cell-cell adhesion dynamically in response to histamine[41]. However, since genome-wide knockout of *Ctnna2* in mice is associated with neonatal lethality, with most homozygotes dying with 24 h after birth[42], vascular-specific deletion of *Ctnna2* may be necessary to interrogate its potential role in Histh and SCLS. *RAD18/Rad18* is a E3 ubiquitin-protein ligase involved in post-replication repair of UV-damaged DNA. Notably, *Rad18* knockout mice exhibit higher Hct at homeostasis[43]. Finally, *CNTN3/Cntn3* encodes contactin 3, an activator of the small GTPase Arf6 that has been linked to inflammation-triggered vascular permeability[44]. $Cntn3^{-/-}$ mice exhibit normal physiological levels of serum albumin and reduced Hct compared to WT controls[38].

Because several of the genes captured in Histh map to chromosomal locations that are outside of the top region of synteny on Chr3 in humans (Fig. 6a), there may be additional causal loci for SCLS. Indeed, because there were 139 separate loci associated with SCLS in our original GWAS[7], we focused on the strongest genetic signal from the SCLS GWAS that allowed us to interrogate candidates that overlap the *Histh* locus. Even within *Histh*, the true causal variants for SCLS may lie in strong linkage with genotyped SNP. In future studies, it will be critical to further dissect the *Histh* QTL by generating interval-specific recombinant sub-congenic lines[45] and/or by using mouse GWAS[46] in order to fine map and validate the true causative genes. Since the rarity of SCLS in humans poses a large barrier to mapping the causative gene(s), our previously unreported mouse genetic model provides an important alternative approach.

The present results advance the field by showing that SCLS patients are hyper-responsive to mediators of vascular permeability, suggesting that aberrant endothelial function contributes directly to clinical symptoms. Although SCLS patients are typically asymptomatic between episodes, dermal microvascular ECs isolated from a patient with fatal SCLS were persistently hyper-responsive to inflammatory mediators including LPS, TNFα, and IL-1β in vitro[8]. Previous histological studies of skin and muscle of SCLS patients have failed to uncover gross structural or

ultrastructural abnormalities within the microvasculature that could account for this phenotype[2]. Thus, our results demonstrating that the cutaneous vasculature of SCLS patients is hyper-responsive to inducers of permeability supports the hypothesis that the acute manifestations of SCLS result from the exaggerated functional responses of a susceptible host to otherwise common inflammatory triggers in a fashion attributable to underlying genetic defects within the endothelium, resulting in an accelerated breakdown of vascular barrier function. As such, future studies including assessment of expression and sequence of top *Histh* candidate genes in SCLS patients and mice and their role in endothelial responses to inflammation will be essential to determine their contribution to these phenotypes.

## Methods

**Patients and skin testing.** Patients were seen at the Clinical Center of the National Institutes of Health under an IRB-approved study protocol (I-0184) after informed consent. Histamine phosphate or morphine sulfate were injected intradermally at separate sites along the dorsal aspect of the upper arm. After 15–20 min, the size of wheals was determined manually and analyzed using ImageJ.

**Animals.** B10.S-*Histh*[SJL], (B10.S × SJL/J)F$_1$ and (B10.S × SJL) F$_2$ and (B10.S × SJL/ J) × B10.S of the Given Medical Building at the University of Vermont according to National Institutes of Health guidelines. Approximately equal numbers of male and female mice were used for each experiment, and ages of mice used are indicated in the accompanying figure or table legend. All animal studies were approved by the Institutional Animal Care and Use Committee of the University of Vermont or the NIAID/NIH (animal study protocol LAD3E).

**Histh phenotyping.** Cohorts of four male and four female mice were used in each histamine challenge. Each mouse was sensitized (day 0 and day 7) by subcutaneous injection with a 50/50 mix of CFA and PBS or left unmanipulated. 30 days later histamine hypersensitivity was determined by intravenous (i.v.) injection of 100, 50, 25, and 6.25 mg/kg histamine (dry weight free base) diluted in PBS. Deaths were recorded at 30 min post-injection and the data reported as the number of dead mice over the number of mice in the study. 6-month old/aged animals did not receive any CFA priming prior to histamine challenge.

**DNA extraction and genotyping.** DNA was isolated from mouse tail clippings as previously described[47]. Briefly, individual tail clippings were incubated with cell lysis buffer (125 μg/ml proteinase K, 100 mM NaCl, 1 0 mM Tris-HCl (pH 8.3), 10 mM EDTA, 100 mM KCl, 0.50% SDS, 300 μl) overnight at 55 °C. The next day, 6 M NaCl (150 μl) was added followed by centrifugation for 10 min. at 4 °C. The supernatant layer was transferred to a fresh tube containing 300 μl isopropanol. After centrifuging for 2 min, the supernatant was discarded, and the pellet washed with 70% ethanol. After a final 2 min. centrifugation, the supernatant was discarded, and DNA was air dried and resuspended inl TE. Genotyping was performed by using established microsatellite markers[17]. Polymorphic microsatellites were selected to have a minimum polymorphism of 8 bp for optimal identification by agarose gel electrophoresis. Briefly, primers were synthesized by IDT-DNA (Coralville, IA) and diluted to a concentration of 10 μM. PCR amplification was performed using Promega GoTaq according standard methods and amplicons were subjected to 2% agarose gel electrophoresis and visualized by ethidium bromide and UV light. All primer sequences are shown in Supplementary Data 3.

**Data resources in the Mouse Phenome Database.** Phenotype data for systolic blood pressure (MPD#23602), hematocrit (MPD#31825) and albumin (MPD#24451) were queried using the Mouse Phenome Database (https://phenome.jax.org/) for laboratory inbred strains. The significance of the observed differences for each trait was determined using the Mann-Whitney U test comparing SJL/J against the mean trait variables for all strains studied.

**Miles assay.** To assess histamine-induced vascular leak in mice, we used the Miles assay as described previously[26]. Briefly, mice were injected intraperitoneally with pyrilamine maleate (4 mg per kg body weight, Sigma) 30 min prior to injection with EB dye to reduce background permeability during handling. Mice were then injected with 100 μl of 0.5% EB dye in PBS (Sigma) via retro-orbital injection, followed by intradermal injections of histamine or saline (50 μl total volume). 30 min after the intradermal injection, the dorsal skin was collected with a 12-mm biopsy punch, and EB dye was extracted with formamide (Sigma; 56 °C for 48 h). The amount of EB in each sample was determined by measuring the absorbance at 620 nm, and results were expressed as EB dye amount (ng) per 100 mm$^2$ of the skin, with quantification against a standard curve.

**Phenotypic analysis of SJL and B10.S**. Blood was collected by tail vein bleed in untreated mice. Mice were treated with histamine (25 mg/kg) by the intraperitoneal route and then anesthetized using isoflurane at $t = 10$ min. Blood was obtained by cardiac puncture and analyzed by the NCI Pathology and Histology Laboratory. Organs we collected and fixed in neutral buffered formalin. Paraffin-embedded tissue sections were stained with hematoxylin and eosin and examined by microcopy.

**Influenza virus infection**. Mice were anesthetized with isoflurane and influenza A/HK/1/68 (H3N2) virus was administered intranasally in a total volume 3 μl in each nare ($6 \times 10^4$ cfu/ml) under Biosafety Level 2 conditions. Weight loss was monitored, and mice were sacrificed on day 8 post-inoculation for analysis of vascular leakage.

**Systemic vascular leak analysis**. To assess influenza-mediated vascular leak in various tissues/organs, we injected mice intravenously with 100 μl of 2% EB in PBS retro-orbitally. Fifteen minutes post-injection, the mice were deeply anesthetized by isoflurane inhalation and perfused with 5 ml of heparinized PBS through the left ventricle of the heart to remove the EB remaining in the vascular space. Tissues were heated at 95 °C for 1 h to obtain dry weights. The amounts of EB dye (ng) were quantified as described above and normalized by dry weights of individual tissues (mg). Results were expressed as fold change compared to corresponding controls (PBS inoculated mice). To analyze histamine-mediated systemic vascular leak, same procedures were performed except that mice were injected with EB dye immediately following intraperitoneal injection with 100 μl of histamine in PBS (12.5 mg per kg body weight).

**Linkage analysis and generation of *Histh* congenic mice**. Segregation of genotype frequency differences with susceptibility and resistance to Histh in (B10.S × SJL) F[2] and (B10.S × SJL/J) × B10.S were tested by Chi-square ($X^2$) analysis in 3 × 2 and 2 × 2 contingency tables, respectively. B10.S-*Histh*[SJL] congenic mice were derived by marker-assisted selection of SJL/J derived alleles and successive backcrossing to B10.S mice.

**Synteny mapping between *Histh* locus and SCLS GWAS candidates**. All 653 SNPs with significant association to SCLS ($p < 10^{-3}$) were retrieved[7] and annotated to 139 human genes using SNP Nexus tool[48]. Using batch query function of MGI-gene database at Jackson Labs (http://www.informatics.jax.org/batch), we retrieved orthologous mouse genes and their genomic coordinates. Genes that mapped outside the *Histh* interval were excluded from subsequent analysis. This analysis yielded nine genes. To test the possibility that the causal gene for SCLS is mechanistically involved in the phenotype, we constructed protein functional interaction network of shared sub-phenotypes including aging, inflammation, vascular permeability and anaphylaxis. We used Gene Weaver[49] to identify genes associated with each term. On the Gene Weaver homepage (https://geneweaver.org), we entered each term. We restricted the search to human, rat, and mouse genes, and only to curated lists. In addition, we used Gene Expression Omnibus (GEO) and PubMed to retrieve expression datasets for each of the 4 phenotype terms. The datasets included GS207241, GS304268, GS232609, GS307787, GS307786, GS187073, GS329405, GS303836, GS306551, GS177812, GS321902, GS335624, GS200192, GS326395, GS326394, GS314384, GS198727, GS338376, GS312527, GS331321, GS189720, GS305542, GS336659, GS208205, GS203935, GS326401, GS326400, GS198840, GS328380, GS331633, GS196333, GS181559, GS200642, GS309839, GS188347, GS303399, GS335186, GS188343, GS303855, GS303854, GS332534, GS185831, GS310883, GS312925, GS206597, GS333133, GS333132, GS184737, GS178946, GS200635, GS200634, GS325136, GS179569, GS305809, GS179567, GS336539, GS197625, GS310812, GS311831, GS333694, GS313895, GS314765, GS307489, GS314761, GS333693, GS306500, GS305074, GS305689, GS183207, GS195522, GS187312, GS181934, GS183208, GS202307, GS178766, GS337531, GS319718, GS180832, GS331235, GS194554, GS309840, GS202668, GS338380, GS334637, GS337325, GS200006, GS336285, GS334132, GS314766, GS309282, GS315491, GS189767, GS197979, GS304020, GS329747, GS309281, GS198350, GS329599, GS338375, GS304858, GS195703, GS304856, GS314760, GS338379, GS244492, GS333363, GS329214, GS337237, GS182049, GS190346, GS181190, GS208837, GS324767, GS183834, GS311371, GS305333, GS313571, GS232381, GS332656, GS332654, GS334710, GS185484, GS310278, GS198753, GS338284, GS190449, GS335383, GS183398, GS318057, GS317572, GS313665, GS205200, GS307505, GS309512, GS191287, GS328379, GS328378, GS331322, GS303853, GS244262, GS311369, GS318901, GS191896, GS308664, GS312985, GS319230, GS320240. We then downloaded all gene sets associated with each term and compiled a final list by removing all duplicates. Mouse homologs for each gene were retrieved using the batch query function in the MGI-gene database at Jackson Labs (http://www.informatics.jax.org/batch). Functional networks for each phenotype term were generated using Ingenuity Pathway Analysis (Quiagen Inc., https://www.qiagenbioinformatics.com/products/ingenuity-pathway-analysis) and overlaid with SCLS GWAS candidates that overlap *Histh*. Biological interactions were predicted using the connect tool and default settings.

**Statistics and reproducibility**. Unless otherwise noted, experiments were repeated at least 3 times, and data are presented as mean ± s.e.m. Data were analyzed using *t* test or ANOVA by PRISM (GraphPad) as indicated in the figure legends. *p* values < 0.05 were considered statistically significant.

**Reporting summary**. Further information on research design is available in the Nature Research Reporting Summary linked to this article.

## Data availability

All data generated or analysed during this study are included in this published article, Supplementary Information, and Supplementary Data. Genetic linkage of mouse genome with histamine hypersensitivity (Histh) is shown in Supplementary Data 1. The Source data underlying the plots shown in Figs. 1–6 are provided in Supplementary Data 2. The accession number for the Histh trait in the mouse genome informatics database (www.informatics.jax.org) is MGI:6360897. This trait can be accessed under the name "Histh". SNP data were retrieved from publicly available databases (mouse phenome database, https://phenome.jax.org/snp/retrievals; Mouse genomes project,https://www.sanger.ac.uk/sanger/Mouse_SnpViewer/rel-1505). A list of studies that have been used to generate phenotypic data are available at: https://phenome.jax.org/about/snp_retrievals_help.

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

## Acknowledgements

Funding & Disclaimer: Funding for this study was provided in part by the Division of Intramural Research, National Institute of Allergy and Infectious Diseases, National Institutes of Health (project numbers Z01-AI-000746; Z01-AI-000943).The content of this publication does not necessarily reflect the views or policies of the Department of Health and Human Services, nor does the mention of trade names, commercial products, or organizations imply endorsement by the U.S. Government.

## Author contributions

A.R. and Z.X. performed experiments, analyzed data, and wrote the paper. E.C.C., W.S.C. performed experiments and analyzed data. A.R.E., L.M.S. recruited patients, performed experiments, and edited the paper. D.N.K., H.F.R., S.M.P., E.P.B. provided results, analyzed data and edited the paper. C.T., K.M.D. conceived and supervised the project and wrote the paper.

## Competing Interests

The authors declare no competing interests.
