## [Peer Review File · Communications Biology]

Reviewers' comments:

Reviewer #1 (Remarks to the Author):

The authors selected SJL strain as a model for SCLS based on the predisposition of three major symptoms and histamine sensitivity and determined genetic loci for the disease by linkage analysis. Genetic analyses of rare diseases are quite difficult because of the limited availability of human specimen. The authors overcame such difficulty by the strategy using appropriate model selection. I think the manuscript is well-written and the logic of the study was clear. I would like to propose some ideas for the authors to make the logic of the manuscript more concrete:

1) The authors tested the histamine sensitivity by death rates. The authors described histamine-induced vascular hyperpermeability in SJL mice, but I think the authors need to prove the direct link from the vascular hypersensitivity to death. I am just wondering if there may be other reason(s) for the histamine-induced death. To make this clear, the authors need to describe autopsy reports, like blood parameters like plasma albumin level, and hematocrit of the dead mice and check the similarity to SCLS.

2) I think that the selection of control strains is an important key for linkage analysis. Therefore, I would like the authors to explain in more detail about why B10.S was selected as a control because the figure2A does not contain B10.S strain.

3) Can the authors add more discussion about the involvement of genes shown in Fig 6? Knock out mice for most genes have already been produced and the information on the KO mice can be referred to, e.g., MGI website of the Jackson Laboratory. The authors mentioned the syntenic similarity, but only three out of 9 genes in Fig 6 are mapped in Human 3p25.3. I think the authors can make some comment on this by discussing the functions of the genes listed in Fig 6.

4) Please check minor typing errors, e.g., no label "B" in the legend of Figure 2.

Reviewer #2 (Remarks to the Author):

In this manuscript " A natural mouse model reveals genetic determinants of Systemic Capillary Leak Syndrome (Clarkson Disease)" the authors Raza et al identify an inbred mouse strain SJL that phenomimics the disease particularly the susceptibility to histamine triggered endothelial permeability. They map this histamine hypersensitivity trait to chromosome 6 in the mouse which is syntenic to the genomic locus, 3p25.3 which is strongly associated with SCLS in humans.

This is a well performed study using both human and mouse data. The experiments are well designed with clear read-outs and the data interpreted cautiously and accurately without over reach.

The results of this study will be important not just to understanding SLCS but also for other diseases where systemic breakdown of the endothelial barrier is a significant part of the pathology.

Minor Concerns:

In the discussion (Line 15) the authors state..."although deeper investigation revealed that these mice maintain the prototypic SCLS triad of high hematocrit, low serum albumin and hypotension at hemostasis (Figure 1)". Figure 1 does not depict hematocrit, serum albumin or blood pressure measurements. It seems that the authors have collected this data and therefore should be included in this report.

Reviewer #1 (Remarks to the Author):

The authors selected SJL strain as a model for SCLS based on the predisposition of three major symptoms and histamine sensitivity and determined genetic loci for the disease by linkage analysis. Genetic analyses of rare diseases are quite difficult because of the limited availability of human specimen. The authors overcame such difficulty by the strategy using appropriate model selection. I think the manuscript is well-written and the logic of the study was clear. I would like to propose some ideas for the authors to make the logic of the manuscript more concrete:

1) The authors tested the histamine sensitivity by death rates. The authors described histamine-induced vascular hyperpermeability in SJL mice, but I think the authors need to prove the direct link from the vascular hypersensitivity to death. I am just wondering if there may be other reason(s) for the histamine-induced death. To make this clear, the authors need to describe autopsy reports, like blood parameters like plasma albumin level, and hematocrit of the dead mice and check the similarity to SCLS.

Response: Physiological responses to systemic histamine surges in mice have been studied extensively over the last 70 years, and the results strongly support our rationale to model acute SCLS using this technique. In 1949, McMaster and Kruse first described events following initiation of histamine-mediated shock in mice (PMID:18129860). First, there is a near instantaneous vasoconstriction of arterioles and venules, followed by vasodilation and development of profound hypotension over the next 5-20 minutes. Similar to SCLS, this response is followed by a marked rise in hematocrit associated with a ~30% loss in blood volume due to extravasation of fluid and protein into peripheral tissues (PMID:13465749, 1957; 14276289, 1965). The singular importance of the vascular bed in response to histamine shock was confirmed when it was found that mice could be protected from death by i.v. administration of 6% dextran solution or physiologic saline as blood volume expanders (PMID:14328697,1965). Several other species exhibit signs of vascular leak in internal organs in response to histamine, but none of these are observed in mice (summarized in Munoz, J.J. & Bergman,R.K. (1973) in Immunology Series, ed. Rose, N.(Dekker, NewYork),Vol.4, pp. 1-235). For example, guinea pigs exhibit profound bronchospasm and hypoxemia, while rabbits have acute pulmonary hypertension that is due to vasoconstriction of pulmonary arterioles, in turn leading to acute right ventricular dilation and heart failure. Dogs exhibit severe liver congestion immediately prior to death. In contrast, a mouse examined immediately after death has collapsed

and unobstructed lungs, a heart that is still beating, and an uncongested liver. Except for a generalized edema and erythema of small intestines and stomach, the abdominal organs appear normal. Overall, the phenotype of histamine-mediated shock in mice is strikingly similar to acute SCLS in humans, where edema of visceral organs (lungs, kidneys, liver) is typically absent even when edema of peripheral tissues is present. Histamine shock in the mouse can best be characterized as a combination of hypovolemic shock and low-resistance shock, i.e., significant loss of fluid from the intravascular space and inability to compensate sufficiently to maintain blood pressure and venous return to the heart. These effects can readily be blocked by histamine H₁ receptor antagonists administered prior to or immediately after histamine injection (Ref. 18).

2) I think that the selection of control strains is an important key for linkage analysis. Therefore, I would like the authors to explain in more detail about why B10.S was selected as a control because the figure2A does not contain B10.S strain.

Response: The primary reason we selected B10.S mice as a control strain is that they are congenic for the SJL/J major histocompatibility antigen (*H2*), with both possessing the *H2^s* haplotype. We have included this information in the revised text (see page 5, paragraph 1). In our previous work, we found that controlling for MHC-linked effects on inflammation in mice is paramount (Ref. 18). In these previous studies, we examined genetic determinants of experimental autoimmune encephalomyelitis (EAE), a neuroinflammatory disease resembling multiple sclerosis in humans. Multiple cohorts of backcross (BC1) and F2 intercross mice were generated and immunized with syngeneic mouse spinal cord homogenate emulsified in complete Freund's adjuvant (CFA), a potent inflammatory stimulus. The mapping population used in the current study is an extension of our EAE studies, in which both experimental and control cohorts were challenged with histamine at D30 post-injection. The phenotyped BC1 population used in this study allowed us to perform genome-wide linkage analysis to map the gene(s) controlling histamine hypersensitivity that we had observed in SJL/J mice but not B10.S mice.

Given that neither B10.S nor C57BL/10SgSnJ (the background recipient sub-strain used to generate B10.S) mice have been phenotyped for systolic blood pressure, hematocrit and serum albumin levels, we utilized the data for each of these parameters for genetically highly related strains (C57BL/10, C57BL/6, C57BLKSC57BR, C57L, and C57BR (PMID:15342563)). We

used only data that were obtained simultaneously with SJL/J mice and closely controlled for age, sex, and time of day for specimen collection. These data are included in revised **Figure 2** (page 7) and clearly show that blood pressure, albumin, and Hct in C57BL/10SgSnJ-related strains are significantly different from SJL/J mice.

3) Can the authors add more discussion about the involvement of genes shown in Fig 6? Knock out mice for most genes have already been produced and the information on the KO mice can be referred to, e.g., MGI website of the Jackson Laboratory. The authors mentioned the syntenic similarity, but only three out of 9 genes in Fig 6 are mapped in Human 3p25.3. I think the authors can make some comment on this by discussing the functions of the genes listed in Fig 6.

Response: We revised the figure to make the synteny clearer (**Figure 6a**) and performed analysis of protein function networks (**Figure 6b**). We revised the Results and Discussion to expand our analysis of several gene candidates as outlined below:

Results (see page 16) :

“The extreme rarity of SCLS has greatly limited our understanding of the complex genetic factors that contribute to disease development. The one and only published genome-wide association study of SCLS patients predicted genetic associations (653 SNPs, 139 genes) linked to disease including 3 SNPs in *CAV3* on Chr3p25.3 ($p \sim 10^{-6}$), with an odds ratio of ~ 41 , as the highest-ranking susceptibility locus (Ref 7). Considering the similarity between Hhs and SCLS, we determined whether the *Hhs* locus harbors any of the SCLS GWAS candidates. Synteny mapping revealed several human genes, including *CAV3*, *RAD18*, and *ATP2B2*, that were also captured in *Hhs* (**Figure 6a**). We generated protein functional interaction networks (PMID:24336805) of shared sub-phenotypes between Hhs and SCLS disease to interrogate mechanistic links between *Hhs*-associated genetic loci and disease (**Figure 6b**). This approach identified several genes that are associated with aging (*ATP2B2*, *CAV3*, *CNTN3*, *CTNNA2*, *GRID2*), inflammation (*ATP2B2*, *CAV3*, *RAD18*, *KBTBD8*), vascular permeability (*SFXN5*, *RAD18*) and anaphylaxis (*CAV3*, *RAD18*, *CTNNA2*, *ATP2B2* and *GRID2*). In summary, these results suggest that SJL mice and human subjects with SCLS share a similar genetic basis for increased susceptibility to vascular hyperpermeability.”

Discussion (see page 19-20):

“We further characterized potential functions of several of these genes in SCLS by searching PubMed and the International Mouse Phenotyping Consortium (<http://www.mousephenotype.org>) for associations with processes involved in vascular barrier integrity. Among these, *ATP2B2/Atp2b2* encodes ATPase plasma membrane Ca^{2+} transporting 2 protein, which plays a critical role in intracellular calcium homeostasis and endothelial cell responses to histamine (PMID:10957669). *Atp2b2* also regulates endothelial nitric oxide (NO) synthase (eNOS) phosphorylation in endothelial cells (PMID:20211863), a critical step in histamine- and VEGF-induced vascular permeability previously implicated in SCLS-associated vascular dysregulation (PMID:24046447; 25544902). Notably, *Atp2b2*^{-/-} mice have significantly reduced serum albumin levels at homeostasis compared with WT controls (PMID:27626380). *Cav3* encodes caveolin 3, a protein also implicated in eNOS regulation (PMID:25694588). Although *Cav3* knockout mice develop heart failure due to myocardial fibrosis and dilated cardiomyopathy (PMID:30028203), functions of *Cav3* in the peripheral vasculature have not been studied in mice. *CTNNA2/Ctnna2* encodes α -catenin 2, which functions as a linker between cadherin adhesion receptors and the cytoskeleton, and thereby regulates cell-cell adhesion dynamically in response to histamine (PMID:30185878). However, genome-wide knockout of *Ctnna2* in mice is associated with neonatal lethality, with most homozygotes dying with 24 hours after birth (PMID:15034585); thus, endothelial-specific deletion of *Ctnna2* may be necessary to interrogate its potential role in Hhs and SCLS. *RAD18/Rad18* is a E3 ubiquitin-protein ligase involved in post replication repair of UV-damaged DNA. Notably, *Rad18* knockout mice exhibit higher Hct at homeostasis (PMID:10884424). Finally, *CNTN3/Cntn3* encodes contactin 3, an activator of the small GTPase Arf6 that has been linked to inflammation-triggered vascular permeability (PMID:23143332). *Cntn3*^{-/-} mice exhibit normal physiological levels of serum albumin and reduced Hct compared to WT controls (PMID:27626380).

“Because several of the genes captured in *Hhs* map to chromosomal locations that are outside of the top region of synteny on Chr3 in humans (**Figure 6a**), there may be additional causal loci for SCLS. Indeed, since there were 139 separate loci significantly associated with SCLS in our original GWAS (Ref. 7), we focused on the strongest genetic signal from the SCLS GWAS that allowed us to interrogate candidates that overlap *Hhs* locus. Even within *Hhs*, the true causal variants for SCLS may lie in strong linkage with genotyped SNP. In future studies, it will be critical to further dissect the *Hhs* QTL by generating interval specific recombinant sub-congenic

lines (PMID:12436049) and/or by using mouse GWAS (PMID:23044826) in order to fine map and validate the true causative genes. Since the rarity of SCLS in humans poses a large barrier to mapping the causative gene(s), our new mouse genetic model provides an important alternative approach.”

4) Please check minor typing errors, e.g., no label “B” in the legend of Figure 2.

Response: Corrected.

Reviewer #2 (Remarks to the Author):

In this manuscript “ A natural mouse model reveals genetic determinants of Systemic Capillary Leak Syndrome (Clarkson Disease)” the authors Raza et al identify an inbred mouse strain SJL that pheno-mimics the disease particularly the susceptibility to histamine triggered endothelial permeability. They map this histamine hypersensitivity trait to chromosome 6 in the mouse which is syntenic to the genomic locus, 3p25.3 which is strongly associated with SCLS in humans.

This is a well performed study using both human and mouse data. The experiments are well designed with clear read-outs and the data interpreted cautiously and accurately without over reach.

The results of this study will be important not just to understanding SLCS but also for other diseases where systemic breakdown of the endothelial barrier is a significant part of the pathology.

Minor Concerns:

In the discussion (Line 15) the authors state...”although deeper investigation revealed that these mice maintain the prototypic SCLS triad of high hematocrit, low serum albumin and hypotension at hemostasis (Figure 1)”. Figure 1 does not depict hematocrit, serum albumin or blood pressure measurements. It seems that the authors have collected this data and therefore should be included

in this report.

Response: This should have referred to **Figure 2** and has been corrected.

Reviewers' comments:

Reviewer #1 (Remarks to the Author):

I think the manuscript is good enough for publication, but I wrote below some suggestions to the authors' responses to my previous comments for further possible improvement.

- 1) I understand that a large amount of literature describes the response to histamine treatments as a good indicator in the study of SCLS. Even so, I would like the authors to have demonstrated by showing clinical/pathological observations that the histamine-treated mice in their study died exactly as described in the literature.
- 2) I don't think it so easy to extrapolate characteristics of B10.S mice from those of C57-related strains. Instead, it would be better if the authors had collected actual clinical parameters, e.g., hematocrits, from B10.S and SJL mice and directly compared the parameters between the two strains because the authors actually used both strains in their study.
- 3) I think the manuscript has been improved because the authors discussed the functional aspects of candidate genes by referring to, e.g., the information on KO mice. One thing I am worried about is the new figure for syntenic correlation (Fig. 6a). I am afraid that the current Fig 6a might mislead the readers to think that all the human genes corresponding to Kbtbd8 to Atp2b2 on mouse Chr 6 were located on human 3q25.3 (actually, only the three genes are on 3q25.3). In this sense, I prefer the original (previous) version of Fig 6.

Reviewer #2 (Remarks to the Author):

While it would have been preferable to have a direct comparison of BP, hematocrit and serum albumin with B10.S or C57BL/10SgSnJ the data in Figure 2 shows the propensity of SJL mice to phenocopy the disease relative to other strains. The authors should address this limitation in the discussion if they are unable to perform these measurements in their laboratories.

Reviewer #1 (Remarks to the Author):

I think the manuscript is good enough for publication, but I wrote below some suggestions to the authors' responses to my previous comments for further possible improvement.

1) I understand that a large amount of literature describes the response to histamine treatments as a good indicator in the study of SCLS. Even so, I would like the authors to have demonstrated by showing clinical/pathological observations that the histamine-treated mice in their study died exactly as described in the literature.

Done. These findings are in new **Fig. 3** and described in the text (page 7, lines 174-195):

“Finally, we confirmed that histamine elicited death of SJL mice in a manner consistent with the published literature^{18,19}. Similar to SCLS in humans, there was a rapid onset of hemoconcentration and hypovolemic shock within five to ten minutes of histamine administration. In both SJL and B10.S mice there was a significant increase in Hct over baseline; however, Hct values were significantly higher in SJL mice than in B10.S mice after histamine administration (~66% vs. 57%, $p=0.02$) (**Figure 3a**). By contrast, serum albumin values were normal and equivalent in both strains prior to and immediately after histamine administration (**Figure 3b**). This finding is consistent with the presentation of SCLS flares, in which serum albumin levels are typically normal at initial presentation, followed by a gradual decrease over the following 24-36 hours²⁰. In further accordance with SCLS in humans, mice examined immediately after death had unobstructed lungs, a small, non-dilated heart, an uncongested liver, and grossly normal kidneys and intestines. Likewise, the histological appearance of heart, liver, kidneys, and small intestines was essentially normal in both SJL and B10.S mice (**Figure 3c**). The lungs also appeared to be normal except for the presence of dense peribronchial and perivascular lymphoid aggregates in some SJL mice. We suspect that these represented reticulum cell tumors, which have previously been reported to develop in aged (greater than six months of age) SJL mice²¹. However, these abnormalities were also detected in lungs of untreated mice and were thus unrelated to histamine administration (**Supplementary Figure 1**). Taken together, our findings suggest that histamine caused death of SJL mice by inducing massive fluid extravasation, resulting in the inability to compensate sufficiently to maintain blood pressure and venous return to the heart, a phenotype which reflects the acute presentation of SCLS attacks in humans.”

2) I don't think it so easy to extrapolate characteristics of B10.S mice from those of C57-related strains. Instead, it would be better if the authors had collected actual clinical parameters, e.g., hematocrits, from B10.S and SJL mice and directly compared the parameters between the two strains because the authors actually used both strains in their study.

Done. These findings are featured in **Figs. 2d-e** (baseline) and **Figs. 3a-b** (post-histamine). While SCLS most commonly presents in older (greater than 50 years of age) adults, the phenotypic traits shown in Fig. 2 were studied in young mice (10-12 weeks of age). Therefore, in order to more closely model SCLS, we measured Hct and albumin in aged SJL and B10.S mice (greater than 6 months of age). Similar to the phenotype of SCLS patients, who are asymptomatic between disease exacerbations and have normal lab values, Hct and albumin values were within the normal range in both SJL and B10.S mice at homeostasis (Figs. 2d-e). By contrast, although Hct rose following histamine administration in both strains compared to baseline, post-histamine Hcts were significantly higher in SJL mice than in B10.S mice, reflecting the clinical phenotype observed (hypovolemia, increased blood viscosity due to

hemoconcentration) (Fig. 3a). Post-histamine albumin values (obtained 5-10 minutes after injection) did not differ from baseline (Fig. 3b), a finding that also mirrors the acute SCLS in humans, where albumin does not begin to decline until well into an episode (typically 12-24 hours after onset) (Ref. 20).

3) I think the manuscript has been improved because the authors discussed the functional aspects of candidate genes by referring to, e.g., the information on KO mice. One thing I am worried about is the new figure for syntenic correlation (Fig. 6a). I am afraid that the current Fig 6a might mislead the readers to think that all the human genes corresponding to Kbtbd8 to Atp2b2 on mouse Chr 6 were located on human 3q25.3 (actually, only the three genes are on 3q25.3). In this sense, I prefer the original (previous) version of Fig 6.

Done. We replaced this with the original version of Fig. 6.

Reviewer #2 (Remarks to the Author):

While it would have been preferable to have a direct comparison of BP, hematocrit and serum albumin with B10.S or C57BL/10SgSnJ the data in Figure 2 shows the propensity of SJL mice to phenocopy the disease relative to other strains. The authors should address this limitation in the discussion if they are unable to perform these measurements in their laboratories.

Done. These findings are included in **Figs. 2d-e** (baseline) and **Fig. 3a-b** (post-histamine). We measured these parameters in SJL and B10.S mice directly as outlined above in response to Reviewer 1, comment 2. We discuss these findings in the text, p. 5, lines 132-148.

REVIEWERS' COMMENTS:

Reviewer #1 (Remarks to the Author):

I think the authors have made reasonable revisions. I have no further comment. I can say the manuscript is now ready for publication.

Reviewer #2 (Remarks to the Author):

The authors have addressed all concerns